# Died or Not Dyed: Assessment of Viability and Vitality Dyes on Planktonic Cells and Biofilms from *Candida parapsilosis*

**DOI:** 10.3390/jof10030209

**Published:** 2024-03-11

**Authors:** Betsy Verónica Arévalo-Jaimes, Eduard Torrents

**Affiliations:** 1Bacterial Infections and Antimicrobial Therapies Group, Institute for Bioengineering of Catalonia (IBEC), 08028 Barcelona, Spain; varevalo@ibecbarcelona.eu; 2Microbiology Section, Department of Genetics, Microbiology and Statistics, Faculty of Biology, University of Barcelona, 08028 Barcelona, Spain

**Keywords:** biofilms, microscopy, imaging, amphotericin B, stain-based methods, yeast staining, fluorescence, live and dead

## Abstract

Viability and vitality assays play a crucial role in assessing the effectiveness of novel therapeutic approaches, with stain-based methods providing speed and objectivity. However, their application in yeast research lacks consensus. This study aimed to assess the performance of four common dyes on *C. parapsilosis* planktonic cells as well as sessile cells that form well-structured biofilms (treated and not treated with amphotericin B). Viability assessment employed Syto-9 (S9), thiazole orange (TO), and propidium iodide (PI). Metabolic activity was determined using fluorescein diacetate (FDA) and FUN-1. Calcofluor white (CW) served as the cell visualization control. Viability/vitality percentage of treated samples were calculated for each dye from confocal images and compared to crystal violet and PrestoBlue results. Heterogeneity in fluorescence intensity and permeability issues were observed with S9, TO, and FDA in planktonic cells and biofilms. This variability, influenced by cell morphology, resulted in dye-dependent viability/vitality percentages. Notably, PI and FUN-1 exhibited robust *C. parapsilosis* staining, with FUN-1 vitality results comparable to PrestoBlue. Our finding emphasizes the importance of evaluating dye permeability in yeast species beforehand, incorporating cell visualization controls. An improper dye selection may lead to misinterpreting treatment efficacy.

## 1. Introduction

Fungal diseases caused by yeasts present a significant threat in the field of medicine [1]. The genera *Trichosporon*, *Rhodotorula*, and *Malassezia* are among the several yeasts responsible for superficial and invasive human infections [1]. *Candida* spp. are responsible for the greatest number of fungal infections caused by fungal pathogens [1,2]. *Candida parapsilosis,* in particular, has been identified as the causative agent in approximately 25% of invasive *Candida* infections in several European countries [3]. Its increasing prevalence, coupled with the emergence of antifungal resistance and its ability to form biofilms, highlights the need for developing novel treatment strategies.

In this context, cell viability assays play a crucial role in evaluating treatment efficacy. Determining the percentage of live cells within a population can be achieved using various techniques, with stain-based methods offering speed and objectivity by relying on the dye’s permeability into the cell membrane. These methods also help overcome the challenge of viable but non-culturable cells [4]. However, in certain scenarios, the impact of a treatment cannot be solely assessed by the proportion of live/dead cells. In such cases, the use of dyes that evaluate physiological/metabolic capabilities, known as cell vitality dyes, becomes essential [4]. 

Currently, there are numerous stain-based methods available; however, there is no consensus or established guidelines for their applications in yeast research, particularly concerning *Candida* spp. Therefore, the aim of this study was to evaluate the performance and utility of four commonly used viability and vitality dyes on *C. parapsilosis* planktonic cells and biofilms treated with Amphotericin B (AmB). Significantly different cell staining was observed in *C. parapsilosis* 11103595 depending on the dye used, emphasizing the importance of carefully selecting and using stain-based methods for yeast studies, particularly when evaluating treatment efficacy.

## 2. Results

### 2.1. Viability and Vitality Staining of C. parapsilosis 11103595 Overnight Cultures

In this study, we employed the well-established stain calcofluor white (CW) as a general dye for yeast cell imaging, which in some cases allowed us to identify permeability issues for yeast staining. First, cells from overnight cultures of *C. parapsilosis* 11103595 were stained with the combinations Syto 9 (S9) + propidium iodide (PI) and thiazole orange (TO) + propidium iodide for the viability assessment, and fluorescein diacetate (FDA) and FUN-1 for the vitality assessment (Figure 1), as described in Section 4.4. 

Observations from the viability assessment revealed that the S9 dye was unevenly incorporated into the yeasts, with some cells exhibiting bright staining, while others remained unstained (Figure 1. S9 + PI. See white arrows). Similarly, the TO dye demonstrated irregular staining, with most cells displaying only faint green dye, indicating poor permeability of this dye. The use of PI as an indicator of dead cells (cells with a damaged membrane) yielded consistent results across both dye combinations, with good staining intensity in all cases.

Regarding the metabolic assessment (vitality), FDA staining showed a high background signal, faint staining in the dyed cells, and intense fluorescence within a dead cell (Figure 1. FDA. See white arrow), suggesting minimal metabolic activity. In contrast, FUN-1 staining revealed red cylindrical intravacuolar structures (CIVSs) in most cells, indicating the presence of metabolic activity (Figure 1. FUN-1). Moreover, FUN-1 displayed good permeability inside the cells, with each cell containing green fluorescence (see white arrow pointing to a metabolically inactive cell in Figure 1. FUN-1) or green/red fluorescence (see red arrow pointing to a metabolically active cell with CIVSs in Figure 1. FUN-1).

Control staining with CW allowed for cell visualization regardless of their respective viability and vitality states. Moreover, since this dye is specific for cell wall chitin, alterations in cell morphology after treatments could also be evaluated [5,6].

### 2.2. Total Biomass and Metabolic Activity of C. parapsilosis 11103595 Biofilms Treated with AmB

*C. parapsilosis* 11103595 biofilms formed on silicon coupons for 24 h were treated with AmB at 2.5 μg/mL (MIC_50_ × 10) for 20 h, following the procedure outlined in Section 4.3. Crystal violet assay and PrestoBlue were used, respectively, for total biomass (cell biomass and extracellular matrix) and metabolic activity quantification (Figure 2) to obtain a reference value of treatment effect on biofilms using conventional techniques [7,8]. Our results demonstrate a significant reduction in both total biomass (Figure 2A) and metabolic activity (Figure 2B) by ~23% and ~39%, respectively, compared to the control after a single dose of AmB treatment.

### 2.3. Viability and Vitality Staining of C. parapsilosis 11103595 Biofilms Treated with AmB

Then, we evaluated the efficacy of AmB treatment on *C. parapsilosis* 11103595 biofilms using stain-based methods under the conditions described (see Section 4.4). For each staining condition, we quantified the cell biomass from the confocal images using COMSTAT 2 and compared the mean values of the treated and control biofilms (Figure 3 and Figure 4). Then, to assess the performance of the viability and vitality dyes, we used the data derived from the CW dye, our cell visualization control, and calculated the corrected biomass.

Regarding the viability dyes (Figure 3), we can see that S9 has a poor performance to assess treatment efficacy in *C. parapsilosis* 11103595 biofilms. The S9 + PI row of Figure 3A shows a very significant (*p*-value < 0.0001) reduction (49%) in the live cells biomass in treated samples compared to the control. Notably, most of the stained cells in treated samples had a pseudohyphae morphology. However, once the images are contrasted with the CW staining, we can see these results are not real because unstained cells become visible in the AmB-treated image. Consequently, when biomass is corrected with the values of CW staining, we did not obtain a statistically significant difference in the live cells of treated samples compared to the control.

Similar results were obtained from the TO dye (Figure 3B). The TO + PI row indicates a reduction of 45% in the biomass of live cells (*p*-value < 0.001) from treated samples compared to the control. However, no difference is observed when the live biomass is corrected using the quantification of the CW dye. On the other hand, PI exhibited bright fluorescence and good permeability in both the control and treated samples, regardless of the viability dye used in combination. Additionally, an increase in the biomass of dead cells was observed in the treated samples compared to the control samples (3.4-fold in the S9 + PI condition and 1.6-fold in the TO + PI condition), as expected after the antifungal action. Although not statistically significant, this increase remained after the correction in biomass with the CW dye values (5-fold in the S9 + PI condition and 1.5-fold in the TO + PI condition).

Despite the unsatisfactory performance of the evaluated viability dyes to assess treatment efficacy in *C. parapsilosis* 11103595 biofilms, we calculated the viability percentage of treated and non-treated biofilms using the uncorrected biomass data from S9 + PI and TO + PI conditions. Then, we calculated the change in the viability percentage obtained after treatment, aiming to see the erroneous conclusions that would be derived from the data. As it can be seen in Table 1, S9 reports a reduction of 15% in the viability percentage of *C. parapsilosis* 11103595 biofilms treated with 2.5 µg/mL of AmB, while TO indicates a change 2.5-fold lower (6%). 

Next, we evaluated changes in the biomass of metabolically active cells in *C. parapsilosis* 11103595 biofilms treated with AmB using vitality dyes (Figure 4). We found that control samples stained with FDA exhibited a high background signal and only a few stained cells (Figure 4A). This noise led to an erroneous elevated biomass value of metabolically active cells (23 ± 5 µm^3^/µm^2^). In AmB-treated samples, cells (mostly elongated pseudohyphae) displayed good intensity of staining. The obtained biomass of metabolically active cells in the treated sample was 27 ± 3 µm^3^/µm^2^; however, no comparison could be made due to the lack of a good control. Moreover, the high background staining of the control also influenced the outcome of the biomass correction using the CW data. Therefore, the obtention of reliable data regarding the change in metabolic activity in *C. parapsilosis* 11103595 biofilms treated with AmB using FDA dye was impossible in our study.

In the case of FUN-1, we found red CIVSs in most cells of the control sample, accounting for a biomass of 15.8 ± 2 µm^3^/µm^2^ (Figure 4B). This value was reduced to 10 ± 0.4 µm^3^/µm^2^ in samples treated with AmB. Once we calculated the biomass of cells metabolically active in each condition (cell biomass from red channel/cell biomass from green channel), we saw a reduction of 37% in treated samples compared to the control. Moreover, when the data were corrected using CW quantification data, this difference was maintained (33%).

Finally, we also calculated the vitality percentages of treated and non-treated biofilms using the uncorrected biomass data from FDA and FUN-1 dyes to obtain the change in vitality percentages after treatment (Table 1). In this way, we could see how FDA data lead to the erroneous conclusion of a 17% increase in the metabolic activity of cells from *C. parapsilosis* 11103595 biofilms after treatment with 2.5 µg/mL of AmB. In contrast, FUN-1 data presented a 37% reduction in the metabolic activity of the biofilm cells, a value very similar to the result obtained by the PrestoBlue assay (Figure 2B).

Notably, the presence of filamentation in cells growing in biofilms allowed us to observe differences in dye penetration based on cell morphology that were not appreciable in the planktonic cultures. Overall, when S9 and TO were employed, cytoplasmic staining was more associated with long pseudohyphae. Instead, localized staining, presumably of nucleic acids, observed as multiple points, was associated with short pseudohyphae and blastospores (Figure 3). 

## 3. Discussion

Dyes used for assessing viability rely on permeability differences between live and dead cells, which are indicative of membrane integrity [4]. However, before implementing a dye, it is crucial to evaluate and standardize its performance in the microorganism of interest to avoid the misinterpretation of treatment efficacy in a real-world context. In this study, permeability differences not associated with cell viability/vitality were observed when using S9, TO, and FDA dyes for *C. parapsilosis* 11103595 staining. 

We decided to further test the dyes’ performances in cells growing in biofilms. *C. parapsilosis* is known for its ability to adhere and form biofilms on medical-related devices [9]. Biofilms formed by *C. parapsilosis* are aggregates of blastospores and/or pseudohyphae embedded in an extracellular matrix that protect the cells from external aggressions [9]. Thus, biofilm formation is associated with a higher resistance to antifungals. Considering the threat that *C. parapsilosis* represents for human health, especially for neonates [9], it becomes important to develop new therapeutic approaches. However, this progress need to be accompanied by established methods that allow testing treatment efficacy. 

Most studies that evaluate the efficacy of novel potential antibiofilm agents use standard techniques (crystal violet assay, CFU plate counting, tetrazolium salts, or resazurin-based methods) accompanied by a visualization method for the confirmation and complementation of results. Direct imaging of biofilms allows the assessment of structural changes after treatment, while fluorescence staining permits selective labeling, providing additional information regarding biofilm composition and the viability/vitality of the microorganisms [10]. However, the fast and accurate visualization of these methods provide them with the potential to be used on their own as alternatives of conventional methods. This is the case for the whole slide imaging technique, a promising tool for the diagnosis and antifungal susceptibility evaluation of *Candida* spp. both in planktonic and biofilm states [11,12,13].

Regarding the results obtained in this study, it is important to mention that the performance of the evaluated dyes was similar in planktonic and biofilm cells. This suggest that the extracellular matrix present in the biofilm does not have a significant influence on dye permeability inside the cells. Although, this means that the problems observed with S9, TO, and FDA in the overnight cultures were also present in biofilms. However, S9 and FDA are commonly used in studies of antimicrobial efficacy on the planktonic and biofilm cells of *Candida* spp. without the use of cell visualization controls [14,15,16,17,18,19]. Moreover, TO is used in cell viability analysis by flow cytometry [20,21], where it is easier to miss the presence of permeability issues. Unstained cells after S9 and TO staining become evident after AmB treatment of biofilms. This interference with dye permeability could be the result of yeast adaptations to antimicrobial treatments as reported previously [22,23,24]. For instance, the modulation of sterol composition has been proposed as a mechanism of resistance against AmB in *Candida* spp. [9].

On the contrary, PI showed good staining intensity on planktonic and biofilm cells. Thus, we believed that using PI as an indicator of cell death (compromised membrane integrity) in combination with a total cell visualization marker, such as bright field, or a membrane/cell wall dye, could be a good alternative to evaluate antifungal activity in *C. parapsilosis*. This strategy has been successfully implemented previously by several authors on different *Candida* spp. [25,26,27]. 

In our study, the identification of permeability issues was possible thanks to the incorporation of CW as a control of cell visualization. This agrees with studies that have proven the usefulness of CW for the rapid identification of *Candida* spp. even in clinical samples [28,29]. However, it is important to carefully select the cell visualization control depending on the treatment under evaluation. For instance, CW may not be the best option when the evaluated treatment affects the cell wall or alters its composition. As an example, take the case of N-acetylcysteine that acts on the polysaccharides of the fungal cell wall or caspofungin that can stimulate the chitin synthesis of *Candida* spp., inducing changes in the cell wall structure and paradoxical growth [24,30].

On the other hand, the ability of yeasts to switch between morphologies adds another level of complexity. Variations in dye distribution inside the cell (cytoplasmatic or associated with nucleic acid binding) were observed, with a tendency of uniform staining more present in pseudohyphae than blastospores. We hypothesize that this variation can be associated with differences in the cell wall structure. Although we did not find a study that compares the cell wall composition of blastopore and pseudohyphae in *C. parapsilosis*, cell length has been associated with increasing adhesion, hydrophobicity, and high expression of mannose-rich glycoconjugates in this species [31].

Quantification using stain-based methods relies on the fact that the fluorescence intensity correlates with the amount of fluorophore present, which in turn represents the number of cell structures or cells in a sample [32]. This principle is applicated in standardized methods that evaluate treatment efficacy in a rapid manner using viability/vitality dyes and fluorescence readers. The same fundament is also employed in the bioinformatic quantification of fluorescence intensity from microscopy images. In our study, we used the latter approach to calculate cell biomass (Figure 3 and Figure 4) and the percentages of viability or vitality of the biofilms treated with AmB for each dye (Table 1). In addition, we used the cell biomass measure obtained from the CW dye and corrected the biomass of the evaluated dyes (Figure 3 and Figure 4).

In the case of viability dyes, both showed a reduction in the viability percentage after treatment, which was discredited once the cell biomass values were corrected by the CW measure (Figure 3). Overall, PI dye exhibited a good staining intensity in the control and treated samples, although no statistical difference was observed among them (Figure 3). The reference crystal violet technique showed a decrease of 23% in total biomass (*p*-value < 0.05) after the treatment of *C. parapsilosis* 11103595 biofilms with 2.5 µg/mL of AmB. However, it is important to consider that this reduction not only considers cell biomass but also extracellular matrix biomass. 

On the other hand, the results obtained by the PrestoBlue assay show that the concentration of AmB used in the study has a greater impact on the metabolic activity of biofilms than on their biomass (Figure 2). However, the high background signal of FDA dye in the control samples prevented the reliable measurement of metabolically active biomass (Figure 4A) and vitality percentage change (Table 1). The fact that cells in the treated biofilm samples exhibited good staining intensity without background noise suggest that FDA’s poor performance can be associated with permeability issues. AmB is a widely used antifungal drug that affects the plasma membrane by creating pores and sequestering ergosterol, leading to increased cell permeability [33]. Therefore, we hypothesize that AmB treatment favors FDA staining by increasing cell dye uptake. 

Conversely, the FUN-1 dye showed a good performance in cell staining with a vitality percentage change similar to that obtained with the PrestoBlue assay (Figure 2B and Figure 4B). The biomass correction with the CW measure showed a reduction in metabolic activity very close to the original (33%), indicating a good reliability of the dye on its own. Our results are in agreement with the study of Cho et al. (2023), in which a good performance of FUN-1 combined with CW for the viability assessment of *C. parapsilosis* cells after treatment with tacrolimus was observed [34]. Similarly, Miranda-Cadena et al. (2021) used a commercial alternative (LIVE/DEAD yeast viability kit) composed of FUN-1 and CW to assess the fungicidal and antibiofilm activities of three phytocompounds against *Candida* spp. [35]. 

Different challenges are observed in the assessment of biofilm susceptibility. Although conventional methods are high throughput, these in vitro models poorly represent the in vivo situation, leading to susceptibility data that may disagree with clinical output [36]. Dynamic models allows media flow above the biofilm surface, mimicking better in vivo conditions and providing more reliable outcomes when evaluating antimicrobial compounds [37]. Several of these devices, including microfluidic platforms, are closed and depend on microscopic readings, usually performed with confocal scanning laser microscopy and fluorescent dyes [37]. In these cases, the correct dye selection becomes very important to standardize the techniques before the susceptibility testing of biofilms, because an adequate choice can help in the assessment of new antifungal/antibiofilm agents and in the development of new methods for assessing biofilm susceptibility, both being important requirements to address in the clinical setting.

In conclusion, our results highlight the importance of evaluating dye permeability in the specific species and growth mode of interest, especially when it comes to yeasts with dimorphism. The differences in fluorescence intensity observed among yeast morphologies present in biofilms suggest variations in permeability. Therefore, before utilizing stain-based methods for quantification purposes, we recommend conducting a microscopic evaluation using bright-field images or membrane/cell wall dyes as controls for cell visualization. This approach can reveal a completely different picture of the situation, as demonstrated in the case of AmB-treated biofilms stained with S9 (Figure 3A). Furthermore, we confirm the good performance of PI and FUN-1 dyes for *C. parapsilosis* 11103595 studies.

Although the findings of this article were derived from a single clinical isolate of *C. parapsilosis*, and therefore, these results may not universally apply to other yeasts, they reinforce the central message of the study: the significance of evaluating dye performance in the specific microorganism of interest before practical implementation.

## 4. Materials and Methods

### 4.1. Bacterial Strains and Growing Conditions

This study was conducted using the fungaemia clinical isolate *Candida parapsilosis* 11103595 [38]. To prevent genetic and/or epigenetic changes due to multiple passages, a fresh loopful of the strain was retrieved from a −80 °C stock each week. The recovery process was produced in Yeast Petone Dextrose (YPD) medium, which consisted of 1% yeast extract (Gibco, Carlsbad, CA, USA), 2% meat peptone (Scharlau, Sentmenat, Spain), 2% D-glucose (Fisher Scientific S.L., Madrid, Spain), and 2% of bacteriological agar (Scharlau, Sentmenat, Spain) for solidification when required. Incubation at 30 °C for 24 h was performed.

### 4.2. Biofilm Formation on Silicon Coupons 

Overnight cultures of ~16 h at 200 rpm and 30 °C were centrifugated at 4000 rpm for 5 min and washed twice with Phosphate Buffer Saline 1× (PBS) (Fisher Scientific S.L., Madrid, Spain). The yeast suspensions were adjusted to a final optical density (λ = 550 nm (OD_550_)) of 0.15 in RPMI-1640 with L-glutamine without sodium bicarbonate (Merck Sigma-Aldrich, Madrid, Spain), supplemented with 0.2% D-glucose (referred to as RPMId) and 10% Fetal Bovine Serum (FBS) (Gibco, Carlsbad, CA, USA).

Autoclaved silicon squares (area of 1 cm^2^, thickness: 1.5 mm ± 0.3 mm) (Merefsa, Sant Boi de Llobregat, Spain) were pre-treated with FBS at 37 °C overnight and washed with PBS. The silicon squares were then placed in 24-well cell culture plates (Labclinics, Barcelona, Spain) with 600 μL of the yeast suspension. Adhesion was allowed to occur for 90 min at 37 °C and 60 rpm. Unattached cells were removed by washing with PBS before transferring the silicon to a new 24-well plate containing fresh RPMId medium. The plate was then incubated under the same conditions for a total of 44 h, including growth and treatment periods.

### 4.3. Biofilm Total Biomass and Metabolic Activity

Biofilms formed over 24 h were treated with 2.5 μg/mL of AmB (Gibco, Carlsbad, CA, USA), which corresponds to approximately ten times the MIC_50_ value of the evaluated strain as previously determined in our laboratory (MIC_50_ = 0.25 μg/mL). After an additional 20 h of treatment, total biomass and metabolic activity were measured as described below. Biofilms treated with media alone (RPMId) were used as controls.

Biofilm total biomass was firstly quantified by staining the silicon square with 0.1% (*v*/*v*) crystal violet (Merck Life Science, Darmstadt, Germany) for 5 min, followed by distaining with 33% (*v*/*v*) acetic acid (Gibco, Carlsbad, CA, USA). The different washes and the fixation step were avoided to prevent biofilm detachment. Then, the crystal violet solution was measured by optical density at 570 nm (OD_570_) using a Microplate Spectrophotometer Benchmark Plus (Biorad, Barcelona, Spain). 

To assess metabolic activity, the total biomass from the silicon squares was resuspended in 1 mL of PBS. Each sample underwent a cycle of vortexing for 1 min, followed by ultrasonic bath treatment in a Branson 200 ultrasonic cleaner (Branson Ultrasonics, Brookfield, WI, USA) for 10 min, and another round of vortexing for 1 min. Cells were then harvested and resuspended in a solution of PrestoBlue Cell Viability Reagent (Invitrogen, Carlsbad, CA, USA) in RPMId media at a 1:10 ratio. The suspension was incubated for 3 h at 37ºC in the dark. Fluorescence (λ_Exc_ = 535 nm and λ_em_ = 615 nm) and OD_570_ were measured using a SPARK Multimode microplate reader (Tecan, Männedorf, Switzerland).

### 4.4. Dyes Evaluation

Four different dyes commonly used for yeast staining [16,17,20,34,39,40,41,42,43,44,45], two for cell viability and two for cell vitality, were selected. Additionally, the chitin dye CW (Thermo Fisher Scientific, Carlsbad, CA, USA) at 10 μM was used as the control for cell visualization.

For the viability assessment, the dyes used were S9 (Live/Dead BacLight Bacterial Viability Kit) (Thermo Fisher Scientific, Carlsbad, CA, USA) at 30 μM and TO (Yeast Live-or-Dye Fixable Live/Dead Staining Kit) (Biotium, Fremont, CA, USA) at 10 μM. These green-fluorescent dyes are membrane-permeable and can stain both live and dead cells. They were tested in combination with PI (Live/Dead BacLight Bacterial Viability Kit) (Thermo Fisher Scientific, Carlsbad, CA, USA) at a concentration of 5 μM as a dead cell indicator.

Vitality was assessed using FDA (Merck Sigma-Aldrich, Madrid, Spain) at 10 μg/mL and the yeast-specific FUN-1 dye (Thermo Fisher Scientific, Carlsbad, CA, USA) at 20 μM. FDA becomes fluorescent green in metabolically active cells with intact membranes, as it is hydrolyzed by intracellular esterases. On the other hand, the FUN-1 dye becomes fluorescent green after binding to protein and nucleic acids [46]. Moreover, endogenous biochemical reactions lead to the formation of red CIVSs, allowing the identification of yeast cells with intact membranes and metabolic capacity [46].

The performance evaluation of the dyes was first performed on planktonic cells from overnight cultures. Cells were harvested with a centrifugation step at 4000 rpm for 5 min and washed twice with PBS before staining. Images at a 100× magnification were obtained using an LSM 800 confocal scanning laser microscope (Zeiss, Oberkochen, Germany) and processed using Fiji – Image J software v1.54f [47].

Then, 24 h-formed biofilms were treated with 2.5 μg/mL of AmB or RPMId for 20 h and subsequently stained for analysis. Images at a 60× magnification were obtained and processed in the same way as with planktonic cultures. Biomass quantification of cells was performed using the ImageJ plugin COMSTAT2 [48,49,50]. Corrected biomass of viability dyes was obtained by multiplying the biomass measure of each channel (green live cells and dead live cells) with the biomass measure from the CW channel. Similarly, the corrected biomass of FDA dye was obtained by multiplying the measure of biomass from the green channel (metabolically active cells) with the biomass measure from the CW channel. In the case of FUN-1, the corrected biomass of metabolically active cells was obtained by multiplying the biomass measure of each channel (green and red) with the biomass measure from the CW channel and then calculating the red channel/green channel ratio, following the manufacturer’s instructions [46].

The percentage of viability was calculated according to the following formula: ((Live Cells/(Live Cells + Dead Cells) × 100, where “Live cells” correspond to the biomass measurement (µm^3^/µm^2^) obtained from S9 or TO staining, and “Dead cells” are determined based on the biomass measurement from PI staining. Similarly, the vitality percentage for FDA was derived from the quantified biomass of stained cells, while the vitality percentage for FUN-1 was calculated as the ratio of red biomass to green biomass.

### 4.5. Quantification and Statistical Analysis

Graphics and comparisons between treated and untreated biofilms were conducted using GraphPad Prism v9. Statistical differences were evaluated by an unpaired *t*-test with a significant level of *p* < 0.05.

## Figures and Tables

**Figure 1 jof-10-00209-f001:**
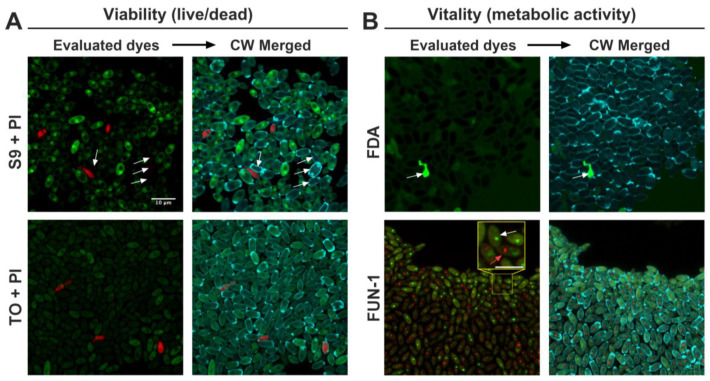
Viability and vitality staining of *C. parapsilosis* 11103595 overnight cultures. (**A**) Viability assessment of *C. parapsilosis* 11103595 planktonic cells using S9 (green) + PI (red) and TO (green) + PI (red) staining. Dyes were compared with CW (blue) merged image for cell visualization control. White arrows in the S9 + PI row highlight some of the cells that were not stained with S9 but were visible with CW. (**B**) Vitality evaluation using FDA (green) and FUN-1 (green and red) staining. CW (blue) merged image was used as a cell visualization control. White arrows in the FDA row indicate high intensity only in a dead cell. White arrow in the zoomed in image of the FUN-1 row shows a green non-metabolically active cell, while the red arrow shows a cell with red cylindrical intravacuolar structures (CIVSs), indicating metabolic activity. Confocal images were processed using ImageJ v1.54f. The scale bar of 10 µm is consistent for all cases, except in the FUN-1 zoomed-in image where it represents a 5 µm length. Images were taken at 100×. S9 = Syto9, PI = propidium iodide, TO = thiazole orange, CW = calcofluor white, FDA = fluorescein diacetate.

**Figure 2 jof-10-00209-f002:**
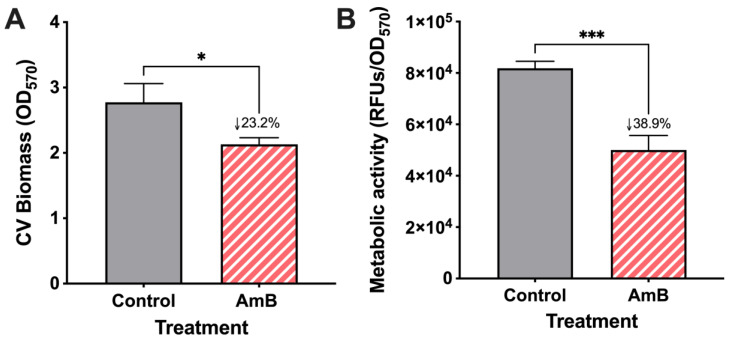
Total biomass and metabolic activity of *C. parapsilosis* 11103595 biofilms treated with amphotericin B (AmB). (**A**) Total biomass quantification (cell biomass and extracellular matrix) with crystal violet (CV) assay and (**B**) metabolic activity evaluation by PrestoBlue assay after 20 h treatment with 2.5 μg/mL of AmB. Biofilm results experiments were conducted in triplicate. Numbers after the symbol ↓ indicate the percentage of decrease in the mean value with respect to the control. Data are represented as mean ± standard deviation. Asterisks indicate statistically significant differences versus control in an unpaired *t*-test (*: *p*-value < 0.05, ***: *p*-value < 0.001). RFUs = relative fluorescence units.

**Figure 3 jof-10-00209-f003:**
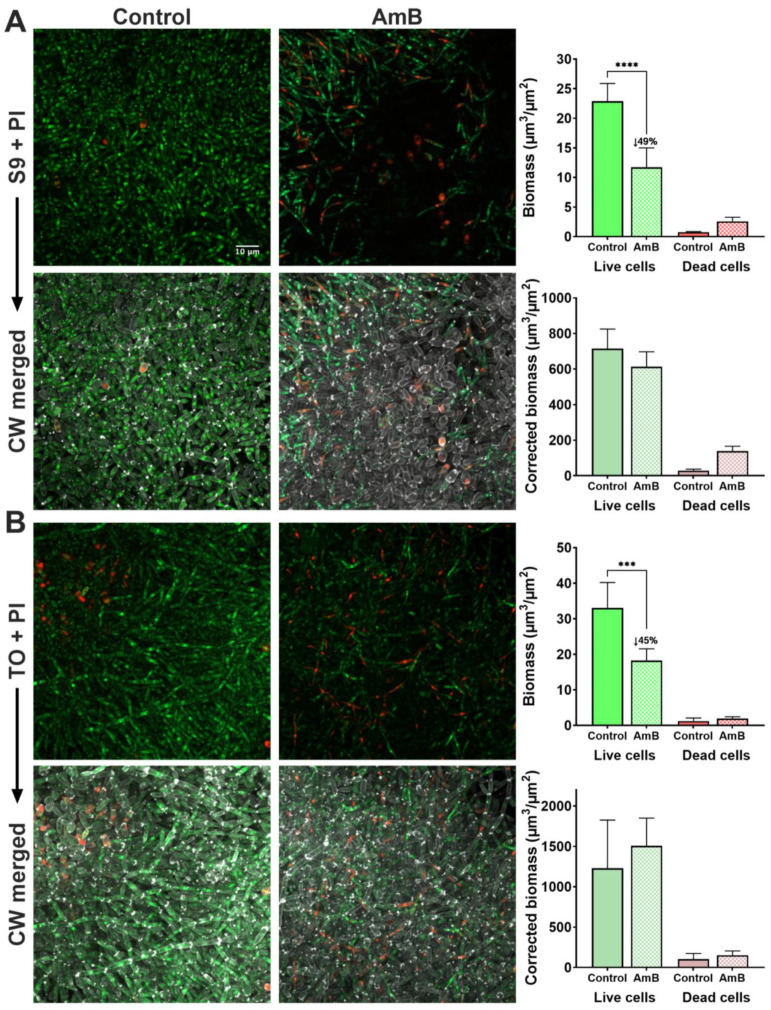
Viability staining of *C. parapsilosis* 11103595 biofilms treated with amphotericin B (AmB). (**A**) Viability assessment using S9 (green live cells) + PI (red dead cells) dye with the respective biomass quantification. (**B**) Viability assessment using TO (green live cells) + PI (red dead cells) dyes with the respective biomass quantification. Each dye combination was compared with the CW (gray) merged image and the respective biomass correction. Biomass quantifications were performed from images at a 10× magnification with the plugin COMSTA2 from ImageJ software v1.54f. Data are represented as mean ± standard deviation from *n* ≥ 3 replicates. Asterisks indicate statistically significant differences versus control (***: *p*-value < 0.001; ****: *p*-value < 0.0001). Numbers after the symbol ↓ indicate the percentage of decrease in the mean value with respect to the control. Z-stack of biofilm top layers from confocal images at a 63× magnification were created using ImageJ v1.54f. The scale bar of 10 µm is consistent for all cases. S9 = Syto9, PI = propidium iodide, TO = thiazole orange, CW = calcofluor white.

**Figure 4 jof-10-00209-f004:**
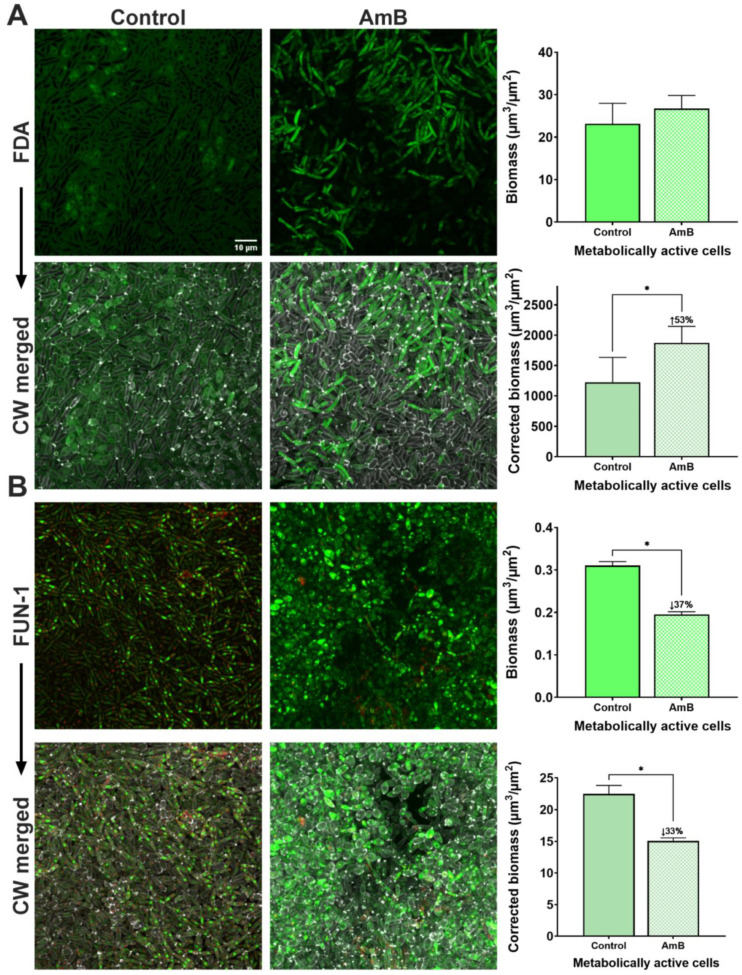
Vitality staining of *C. parapsilosis* 11103595 biofilms treated with amphotericin B (AmB). (**A**) Vitality assessment using FDA (green metabolically active cells) dye with the respective biomass quantification. (**B**) Vitality assessment using FUN-1 (green metabolically unactive cells and green/red metabolically active cells) with the respective biomass quantification. Each dye was compared with the CW (gray) merged image and the respective biomass correction. Biomass quantifications were performed from images at a 10× magnification with the plugin COMSTA2 from ImageJ software v1.54f. Data are represented as mean ± standard deviation from n ≥ 3 replicates. Asterisks indicate statistically significant differences versus control (*: *p*-value < 0.05). Numbers after the symbol ↑ indicate the percentage of increase in the mean value with respect to the control, while numbers after the symbol ↓ indicate the percentage of decrease in the mean value with respect to the control. Z-stack of biofilm top layers from confocal images at a 63× magnification were created using ImageJ v1.54f. The scale bar of 10 μm is consistent for all cases. CW = calcofluor white, FDA = fluorescein diacetate.

**Table 1 jof-10-00209-t001:** Viability and vitality evaluations of *C. parapsilosis* 11103595 biofilms by COMSTAT 2 quantification.

Evaluated Dye	Non-Treated Biofilms	AmB-Treated Biofilms	Change (%)
%Viability (live cells biomass/(live cells biomass + dead biomass))			
S9–PI	96.6 ± 0.3	81.9 ± 2.9	−15.2 ± 2.8
TO–PI	94.9 ± 3.7	89.1 ± 1.4	−6 ± 2.6
Metabolically active biomass (µm^3^/µm^2^)			
FDA	23.1 ± 4.8	26.8 ± 3.1	+17.1 ± 10.5
FUN-1 ^1^	31.1 ± 0.9	19.5 ± 0.6	−37.1 ± 1.1

^1^ Red/green ratio. Data corresponds to mean ± standard deviation from 3 replicates; AmB = amphotericin B; % = percentage; S9 = Syto9; TO = thiazole orange; PI = propidium iodide; FDA = fluorescein diacetate; CW = calcofluor white.

## Data Availability

Dataset available on request from the authors.

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
