# Peer review of "Died or Not Dyed: Assessment of Viability and Vitality Dyes on Planktonic Cells and Biofilms from Candida parapsilosis"

_jof, 2024, doi:10.3390/jof10030209_

Round 1

Reviewer 1 Report

The manuscript is innovative and is well written.

However it needs some corrections that are detailed below:

Major concern

The Results section should be more accurate.

Lines 132-134: you say: “there was a decrease in the proportion of cells stained with S9 and an increase in the proportion of cells stained with TO compared to the control samples”. Please give numerical values and statistical significance. The same in line 136: “a slight increase (?) in PI staining”

Lines 137, 138: “which resulted in an increased number of cell deaths”.?? Query: How much it increased?

Line 152: you say” only a few stained cells”. Query: Which percentage?

Lines 152, 153: FUN-1 152 staining showed the presence of CIVS in ‘most’ cells??

Lines 156, 157: indicating a ‘higher number of metabolically active cells’ in the treated  samples compared to the control samples??

Line 157, 158: FUN-1 staining showed a ‘decreased presence’ of CIVS?

Please be precise in the whole Results section. 

Title: My suggestion is to eliminate the phrase: “Died or not dyed” from the title.

ABSTRACT

4th line: change  “C. parapsilosis cultures and biofilms” to “C. parapsilosis planktonic cells as well as sessile cells that form well-structured biofilms”

INTRODUCTION

1st line: you say “Fungal diseases present a significant threat in the field of medicine [1].”. However, the reference [1] refers only to yeasts. [1] Sharma, M.; Chakrabarti, A. Candidiasis and other emerging yeasts. Curr. Fungal Infect. Rep. 2023, 17, 15-24. Correct the sentence or change to a proper reference.

Lines 29-31: change “Among the yeasts responsible for superficial and invasive human infections are the genera Trichosporon, Rhodotorula and Malassezia [1]” to “The genera Trichosporon, Rhodotorula and Malassezia are among the several yeasts responsible for superficial and invasive human infections [1].

Line 31, 32: change “However, Candida spp. stand out as the most common fungal pathogens in humans [1,2]” to “Candida spp. are responsible for the greatest number of fungal infections caused by fungal pathogens [1,2].”

LInes 332, 33: eliminate “Candidemia has been reported as the first leading cause of healthcare-associated bloodstream infection in certain centers [1]”

Line 50: eliminate the comma “and biofilms”

Line 56: clarify each abbreviation in its first mentioning. Here “CW”. Please note that the Abstract and the main text, each works on its own. So, although you have clarified CW in the Abstract, you must clarify it again in the main text. The same in lines 58, 59: ON, S9-PI, TO- 58 PI, FDA, FUN-1 must be clarified. Revise the whole text.

Line 93: You say “as a proof-of-concept” Do you mean “as a positive control”?

Revise the whole text where you used the expression “proof of concept”. Revise the definition of proof of concept and check if you are using this expression in a proper way.

Line 97: after “biofilms” add a reference.

Line 116: write “viability”  in low-case first letter.

Line 126: change “planktonic cultures” to “planktonic cells”

Line 164: change “displayed” to ”showed”

Line 214: change “represent” to “represents”

Lines 248, 257: change the unit “rpm” to “g” here and throughout the text.

Line 265: change “hours” to “h” here and throughout the text.

My recommendations is major revision.

Legend for Figure 1 and all legends for Figures: Each Figure must work on its own. So, you must clarify all abbreviations in each Figure

Author Response

See enclosed file

Reviewer 2 Report

The presented article is very interesting and the comparison of the efficiency of different fluorescent dyes is very beneficial for further studies. However, there are inconsistencies that need to be fixed, which are detailed below. For the whole article I recommend - It would be better to use only established abbreviations. It is unnecessary to use ON for overnight culture and SS for silicone squares. The full description of C. parapsilosis, i.e. C. parapsilosis 11103595, should be used throughout the text. This is an article created on the basis of the observation of a single isolate, without the full label the information is misleading and generalized. The publication needs to be reworked, and larger magnifications should be given for the images, where both the changes in morphology and in the claims about coloring will be more visible. At this magnification, the claims made by the authors are more than questionable. The whole discussion needs to be redone. It is not possible to discuss the results of the observation of C. parapsilosis and bacteria. There are 19 publications in WOS dealing with C. parapsilosis, biofilm and fluorescence. And a total of 804 publications devoted to C. parapsilosis and biofilm. More detailed comments are below.

In chapter 4.2. is stated The plate was then incubated under the same conditions for a total of 44 h until biofilm development, in chapter 4.3. is then 24 h formed biofilms.. so the value 44 h is a typo?

In contrast, FUN-1 staining revealed that most cells exhibited active metabolic activity, evident from the presence of CIVS (Figure 1. FUN-1. See white arrows in red dots), with a smaller number of non-metabolically active cells (Figure 1. See red arrows). -  It is not clear on what basis this information is based. Even when the image is enlarged, no described difference is visible in the cells marked with a red arrow and a white arrow (from the other red colored cells).

Moreover, since this dye is specific for cell wall chitin, alterations in cell morphology after treatments could also be evaluated. – this is certainly not possible at this magnification and resolution.

We employed the Crystal Violet 94 (CV) assay for biomass quantification – CV colors biomass and EPS (exopolymeric substances), not only biomass

Presto Blue (PB) assay for evaluating metabolic activity - why was MTT or XTT not chosen, which are commonly used to evaluate metabolic activity?

Fig3 White arrows in the FUN-1 column show CIVS in metabolically active cells. -  Even with a large magnification of the images, this statement cannot be detected

The staining variability of S9 and TO were more pronounced when comparing blastospores to pseudohyphae, with brighter staining in the latter.  Can't agree. Because according to this statement, it would clearly follow from the figure that the presence of AmB increased the number of cells, which does not follow from the comparison with CW. Although there seems to be some error in the experiment because the amount of cells in AmB is higher than in control.

In AmB treated samples, we observed the opposite behavior. There was an increased staining of pseudohyphae using FDA, indicating a higher number of metabolically active cells in the treated samples compared to the control samples. -  This claim is questionable. There is no increase in the metabolic activity of the cells. AmB breaks the cell wall and binds to ergosterol. The smaller fluorescent staining in the control is probably caused by a larger amount of EPS in the biofilm cells, which does not allow sufficient contact with the cell and thus display its metabolic activity. There may be an increase in metabolic activity if there is a stress response of the cells to AmB. But there is a lack of experiments with more concentrations of AmB.

Variations in cell wall composition between blastospores and hyphae has been described in Candida albicans. -  A different species is studied here. If C. albicans is not represented in the study, it is not relevant to work with this, unless it is documented by your own analysis. C. albicans behaves differently from the point of view of biofilm formation, it also forms mycelium, not only pseudohyphae.

Furthermore, our results are consistent with previous observations of lower fluorescence intensity of S9 and TO in Gram-negative bacteria compared to Gram-positive bacteria [7-9].  You absolutely cannot compare bacterial cells with eukaryotic ones.

The strain-specific fluorescence intensity of S9 was not associated with cell concentrations, nucleic acid content, or G+C content, suggesting that permeability differences between Gram-positive and Gram-negative bacteria may be the underlying cause [7]. - This information does not belong in this article, the study is about yeast not bacteria.

In C. albicans, a cell wall with low chitin content is associated with increased resistance to AmB, rendering filamentous forms more susceptible [11,12]. -  Not true. In the biofilm of both bacteria and especially yeast, the formation of biofilm (and the associated hyphae formation in yeast) is the cause of increased cell resistance to drugs, up to 1000 times compared to planktonic cells.

Consistent with this, we observed that AmB enhanced TO permeability within pseudohyphae of treated biofilms (Figure 3A. TO + PI), but we did not observe any improvement in S9 permeability after AmB treatment. This cannot be claimed. Antibiotics or antifungal substances can cause changes in the amount of EPS produced in the biofilm. This can be detected by parallel determination of the metabolic activity of the cells using MTT or XTT together with CV (which stains both cells and EPS).

This phenomenon had been previously documented in bacterial studies by incorporating EDTA into S9 and TO staining solution [7,9]. EDTA affects the outer membrane of Gram-negative bacteria by releasing lipopolysaccharides, facilitating the transport of the dye inside the cells [7].  Cannot be compared with the study of bacterial cultures.

Furthermore, the presence of CIVs allowed for clear distinction of metabolically active cells in samples stained with FUN-1 (Figure 1 and 3. See white arrows). -  Not demonstrable, especially at this magnification. And only in the places marked with a white arrow? Isn't it elsewhere? Why is this not quantified using image analysis?

Author Response

See enclosed document

Round 2

Reviewer 1 Report

The paper is a good contribution for the use of 4 dyes on C. parapsilopsis, with the aim to arrive to a consensus to their use and interpretation. Their findings emphasize the importance of evaluating dye permeability in yeast species beforehand,

The manuscript was corrected as required. The authors explain with good arguments why they did not change the title as requested. I accept  the explanation

Author Response

Thank you for all your suggestions along with the revisions.

Reviewer 2 Report

I thank authors for the answers to the questions. Editing the images greatly contributed to the clarity and quality of the obtained results. Unfortunately, even so, the authors did not avoid some mistakes. In the abstract and in the conclusion, the information actually found by them is missing - only cell staining cannot be used for a separate evaluation of the effectiveness of the treatment. It is always necessary to use a different method to determine at least the metabolic activity of the cells. Unfortunately, I cannot agree with the discussion. The discussion starts on line 265 and ends on line 425. Only one discussion is added in the whole text - Branco et al. 2023. This is more of an evaluation of the results than a discussion. It is necessary to discuss the work with other works devoted to Candida sp., preferably C. parapsilosis. There are many of them.

I thank authors for the answers to the questions. Editing the images greatly contributed to the clarity and quality of the obtained results. Unfortunately, even so, the authors did not avoid some mistakes. In the abstract and in the conclusion, the information actually found by them is missing - only cell staining cannot be used for a separate evaluation of the effectiveness of the treatment. It is always necessary to use a different method to determine at least the metabolic activity of the cells. Unfortunately, I cannot agree with the discussion. The discussion starts on line 265 and ends on line 425. Only one discussion is added in the whole text - Branco et al. 2023. This is more of an evaluation of the results than a discussion. It is necessary to discuss the work with other works devoted to Candida sp., preferably C. parapsilosis. There are many of them.

Author Response

Enclosed please find the answer.

Round 3

Reviewer 2 Report

I thank the authors for making changes in the discussion part. The article is very nice.

I have no further comments. All comments have been incorporated into the text.